# Programmable black phosphorus image sensor for broadband optoelectronic edge computing

Seokhyeong Lee[1,3], Ruoming Peng [1,3✉], Changming Wu[1] & Mo Li [1,2✉]

Image sensors with internal computing capability enable in-sensor computing that can significantly reduce the communication latency and power consumption for machine vision in distributed systems and robotics. Two-dimensional semiconductors have many advantages in realizing such intelligent vision sensors because of their tunable electrical and optical properties and amenability for heterogeneous integration. Here, we report a multifunctional infrared image sensor based on an array of black phosphorous programmable photo-transistors (bP-PPT). By controlling the stored charges in the gate dielectric layers electrically and optically, the bP-PPT's electrical conductance and photoresponsivity can be locally or remotely programmed with 5-bit precision to implement an in-sensor convolutional neural network (CNN). The sensor array can receive optical images transmitted over a broad spectral range in the infrared and perform inference computation to process and recognize the images with 92% accuracy. The demonstrated bP image sensor array can be scaled up to build a more complex vision-sensory neural network, which will find many promising applications for distributed and remote multispectral sensing.

[1] Department of Electrical and Computer Engineering, University of Washington, Seattle, WA 98195, USA. [2] Department of Physics, University of Washington, Seattle, WA 98195, USA. [3] These authors contributed equally: Seokhyeong Lee, Ruoming Peng. ✉email: pruoming@uw.edu; moli96@uw.edu

2D semiconductors have many promising potentials in optoelectronics because they afford a wide range of bandgaps with tunable optoelectronic properties[1–3]. Being atomically thin and transferable, they are amenable to heterogeneous integration with photonic circuits and microelectronics to realize advanced functionalities[4–9]. Among 2D semiconductors, black phosphorus (bP) stands out for its tunable bandgap that corresponds to a wide infrared spectral range. Discrete, array, and waveguide-integrated bP photodetectors with compelling performance have been demonstrated for the infrared[10–19]. Leveraging its broadband infrared responses, arrays of bP photodetectors can be utilized for multispectral imaging, which acquires spatial images with spectral information[17,18]. Multispectral imaging combined with artificial neural networks (ANN) has become a powerful tool for biomedical imaging[20–23], fresh food classification[24–26], and surface damage detection on industrial sites[27–29]. This imaging technique generates a tremendous amount of data and consequently is computation-intensive and latency-sensitive, and thus can benefit from the emerging scheme of edge computing[9,30–33]. Preprocessing the images within the sensors at the edge rather than in the cloud can largely alleviate the data streaming load to the servers, improving the bandwidth budget[34–37] and reducing latency and power consumption. These advantages of edge computing have urged the development of optoelectronic edge sensors that combine vision-sensory and computational functionalities in the same devices[8–10,19], which recently have been demonstrated using 2D materials for visible/UV spectral imaging. Realizing such a scheme using bP will extend it to the infrared spectral range, enabling intelligent night vision and multispectral sensing.

Here, we present a multifunctional image sensor that combines the functions of multispectral imaging and analog in-memory computing to implement an in-sensor ANN for image recognition. The image sensor is based on an array of programmable phototransistors made of few-layer bP (bP-PPTs), which are sensitive to a broad infrared spectral range from 1.5 to 3.1 μm in wavelength. The bP-PPT's programmability and memory stem from the stored charges in the rationally designed stack of gate dielectrics that have a long retention time and effectively modulate the conductance and photoresponsivity of the bP channel. The sensor can be programmed and read out both electrically and optically, enabling optoelectronic in-sensor computing, electronic in-memory computing, and optical remote programming, all in one device. It is used as an optical frontend that captures multispectral images in the infrared and performs image processing and classification tasks, demonstrating its promise for distributed and remote sensing applications on household, farming, and industrial sites.

## Results

Figure 1a illustrates the core functions of the bP-PPT array, where the multispectral image in IR range is detected by the remotely programmed each pixel of the array, performing in-sensor computing and subsequent electronic in-memory computing for classification task. Figure 1b depicts the structure of a single bP-PPT device, which consists of a few-layer bP flake as the channel, a stack of $Al_2O_3/HfO_2/Al_2O_3$ (AHA) as the gate dielectric and charge storage layer[38–40], and a top gate electrode. Contrary to the conventional floating gate devices that trap charges on an isolated metallic gate, the bP-PPT stores charges in the $HfO_2$ dielectric layer with a high density of charge trapping sites at ~1.25 eV below the conduction band[41], which offers more reliable and faster operation and simplifies the fabrication process[40,42–44]. For optical access to the bP-PPT, indium-tin-oxide (ITO) is used as the transparent top gate electrode. Figure 1d depicts the band

alignment of the multiple layers in the bP-PPT device. We design the layer structure and select the materials such that charges (electrons or holes) can tunnel from the bP channel through the thin $Al_2O_3$ barrier layer to be stored in the $HfO_2$ layer and effectively modulate the bP channel with field effect. The electron affinity difference between $Al_2O_3$ and $HfO_2$ ($\chi_{Al_2O_3} = 1.5\,eV$, $\chi_{HfO_2} = 2.5\,eV$) provides a high tunneling barrier that facilitates the retention of trapped charges. Moreover, the energy difference between the stored charges and the conduction band of $HfO_2$ determines the storing energy to be ~1.25 eV, enabling optical control of the stored charges—they can be removed by illuminating with visible light (λ < 0.992 μm) but not with infrared light in the telecommunication band or of longer wavelength. The conductance and photoresponsivity of the bP-PPT are controlled by the density of the trapped charges[13,15,45], thus can be set either by applying electrical gate voltage or by shining visible optical pulses, enabling local and remote programming of the device.

To construct an image sensor and processor, we fabricated a 4 × 3 array of bP-PPTs on a single bP flake in a region with a uniform thickness of 11 nm, as shown in Fig. 1c (see Methods for more details about the fabrication process). We optimized the thickness and the deposition process of the AHA multilayer to realize a high charge storage density of $1.8 \times 10^{13}\,cm^{-2}$ (see Supplementary Note 2 for the method used to determine the density). Figure 1e shows the collective measurement results of the source-drain current ($I_{ds}$) of the devices in the array when the gate voltage ($V_G$) is swept. Because charges are injected and stored in the AHA multilayer during the $V_G$ sweep, the $I_{ds}$–$V_G$ curve shows a hysteresis loop with a large memory window of 25 V in $V_G$. The high charge storage density leads to effective control in the electrical conductance of the bP-PPT, achieving an on/off ratio > 200[39]. Since the array is fabricated on the same bP flake, it has excellent uniformity that inter-device variation in the on/off ratio among 9 devices is less than 8% (inset, Fig. 1e).

Figure 2a illustrates the working principle of electrically programming the bP-PTTs by applying voltage pulses to the gate to enable charge tunneling from the bP to the $HfO_2$ layer. The device can first be reset with a depressive pulse (−18 V amplitude, 50 ms duration) to a fully-off state with low conductance. Afterward, it can be programmed by applying positive voltage pulses with amplitude in the range of 10–18 V and a fixed duration of 20 ms (18 V for state #0). By varying the pulse amplitude, the device can be programmed to states of more than 8 distinguishable and stable levels (equivalent to 3 bits) in its conductance when the bP channel is changed from p-type to n-type doping (Supplementary Fig. 1). The tunneling process and the resulting charge density can be modeled with the Fowler–Nordheim tunneling theory (see Supplementary Note 6 for detailed modeling methods). Figure 2c and d show results of four representative states with a long retention time >2000 s (Fig. 2c) and linear $I$–$V$ characteristics (Fig. 2d). The latter is important for its application in analog computing.

Even higher precision can be achieved by programming the devices optically because optical pulses can directly excite the stored charges to remove them (Fig. 2b), and the duration of optical pulses can be controlled more accurately than voltage pulses. We demonstrate optical programming of the bP-PPT devices using optical pulses in the wavelength of 780 nm, which provides sufficient energy to activate the stored charges to overcome the trapping potential (Fig. 1d). Before programming, the bP-PPT is initialized electrically to state #0. Subsequently, it is illuminated with optical pulses with fixed average power (~10 μW at the device) and varying duration so the pulse energy is varied between 10 nJ and 2 μJ. As shown in Fig. 2e, these optical pulses program the bP-PPT to 36 states with different levels of

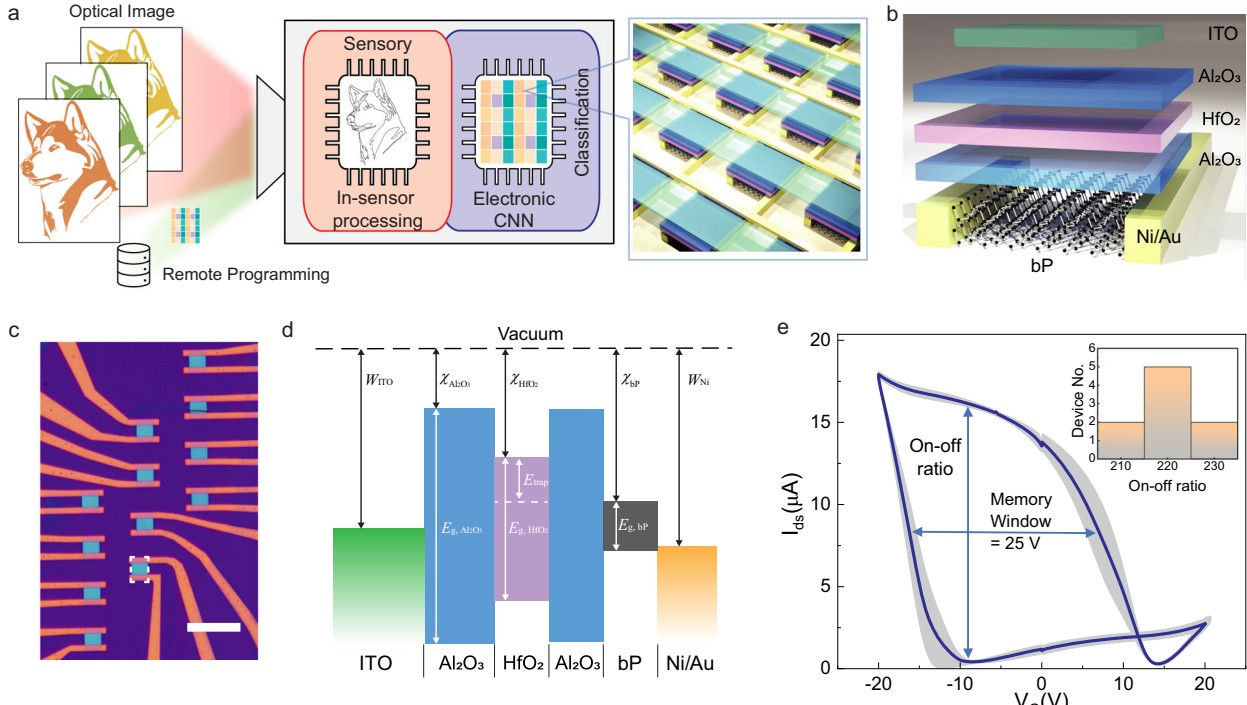

**Fig. 1 Programmable bP phototransistor (bP-PPT) array. a** The bP-PPT array is capable of multispectral infrared imaging and is programmable for in-sensor computing. The array can be programmed remotely using optical control signals and locally using electrical gate voltages. **b** The layer structure of a bP-PPT device. **c** Optical microscope image of a 3 × 4 array of bP-PPT devices patterned on a single bP flake. The white dashed square indicates a patterned bP device. Scale bar: 10 μm. **d** Band diagram of the bP-PPT device. The AHA dielectric stack is designed to store charges in the HfO₂ layer. The electron affinity ($\chi$), bandgap ($E_g$), and work function ($W$) of the layers are: $\chi_{Al_2O_3} = 1.0$ eV, $E_{g,Al_2O_3} = 7.7$ eV, $\chi_{HfO_2} = 2.5$ eV, $E_{g,HfO_2} = 4.9$ eV, $E_{trap} = 1.25$ eV, $\chi_{bP} = 4.4$ eV, $E_{g,bP} = 0.3$ eV, $W_{Ni} = 5.01$ eV, $W_{ITO} = 4.7$ eV. **e** $I_{ds}$–$V_G$ measurement of nine bP-PPT devices. Each bP-PPT shows a large memory window of 25 V in $V_G$ and an on-off ratio of ~200. The inter-device variation (standard deviation) in the array is shown as the gray shaded area. Inset: histogram of the on-off ratio of the devices in the array.

conductance to represent 5 digital bits, a record-high number of levels achieved in charge storage devices. The programming process is accurate, arbitrary, and repeatable. The inset of Fig. 2e shows three adjacent levels that can be programmed repeatedly with high precision. Furthermore, each programmed state is stable for >1000 s, which is shorter than the electrically programmed 8 states but offers higher precision that is sufficient for analog computation applications (Supplementary Fig. 5).

The narrow bandgap of bP enables the bP-PPTs to be operated as broadband photodetectors that can detect optical signals from the near-infrared (NIR) to the mid-infrared (MIR) spectral range. Earlier studies have reported that a bP photodetector's responsivity is sensitive to the doping level and type of the bP channel[12–15,45]. In our bP-PPTs, since we can control the density of the stored charge to modulate the doping level and type of the bP channel, we can program their photoresponsivity in the same way as their electrical conductance. Figure 2f shows the photoresponsivity of a bP-PPT measured in the wavelength range from 1.5 to 3.1 μm when the device is set to high and low conductance states (corresponding to states #0 and #35 in Fig. 2e), respectively. Note that the low conductance state (state #35) has a high photoresponsivity due to the Burstein–Moss effect[12,15]. The unmeasured spectral range (1.8–2.6 μm) is due to the tunability gap of the light source (M-Square Firefly IR). The linearity of the devices' photoresponse is also verified for an incident optical power of up to 30 mW (Supplementary Fig. 3). Therefore, the bP-PPT has a programmable photoresponse in all the telecommunication bands (S, C, and L bands) and the mid-infrared range.

The above results show that the bP-PPT devices can be programmed both electrically and optically. The programmed state is non-volatile and can be read out either electrically by measuring the device's conductance or optically by measuring its photoresponsivity. In both cases, the devices are operated in the linear regime and thus can be utilized for analog computing. Such a hybrid of multifunctional operation modes enables the utilization of a bP-PPT array to implement a mixed-mode optoelectronic neural network system. The same bP-PPT array can act as both the optical frontend to receive and preprocess optical images and an electrical processor with in-memory computing to postprocess the images (Fig. 3a).

**bP-PPT in-sensor convolution for edge detection.** We first use the bP-PPT array to detect infrared optical images and preprocess them in the sensor[8–10,37]. To prove the concept, we configure the bP-PPT array to perform edge detection of images by programming their photoresponsivity (R) to represent convolutional kernel matrices and receiving input images transmitted and encoded in the power ($P_{in}$) of telecom band optical signals. Measuring the photocurrent output $I_{Ph} = R \cdot P_{in}$ from the array corresponds to a multiply-accumulation (MAC) operation[46–48] on the input image with the kernel matrix stored in R. For edge detection, the photoresponsivity matrix R of a 2 × 2 bP-PPT array is optically programmed to binary values (Fig. 2f) and, after proper normalization, to represent kernel matrix $\begin{bmatrix} -1 & 1 \\ -1 & 1 \end{bmatrix}$ for

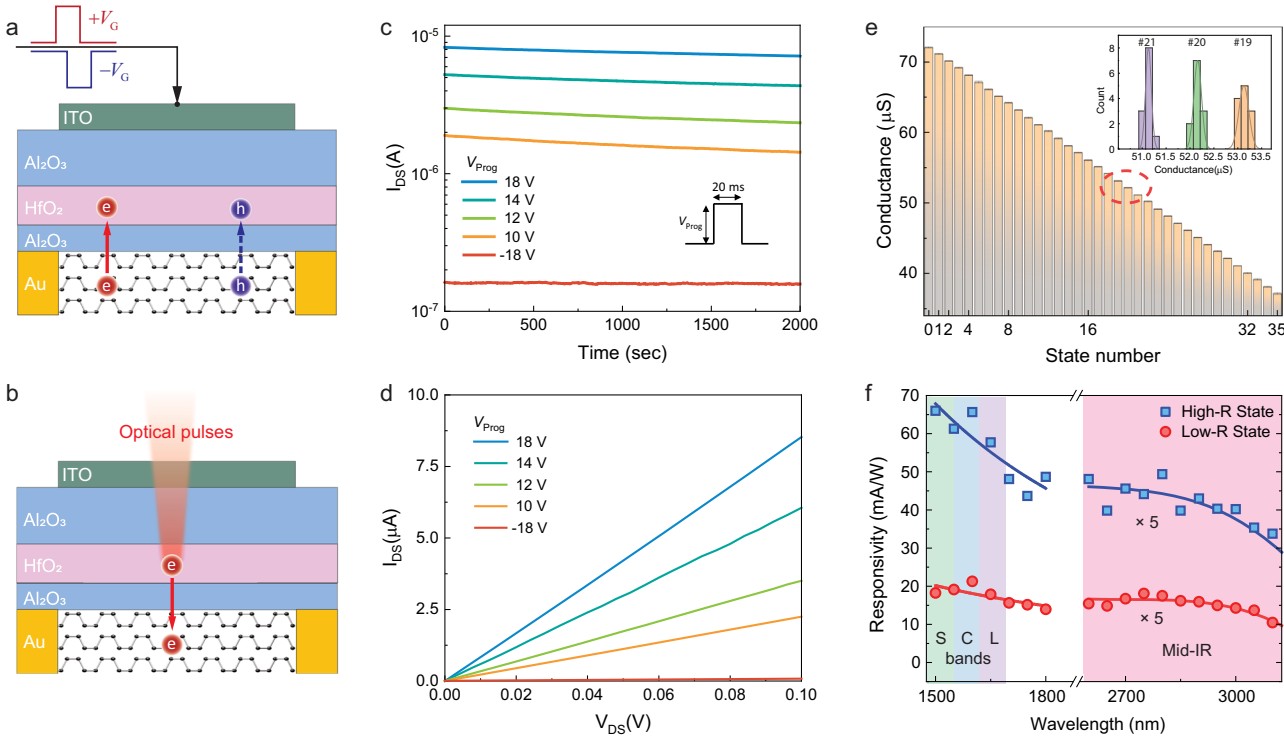

**Fig. 2 Mixed-mode programming and operation of the bP-PPT device.** Schematic illustrations of the working principle of programming the bP-PPT using **a** electrical gate voltage pulses and **b** optical pulses. The red and blue circles represent the electrons and holes in the bP channel or HfO$_2$ trap sites, respectively. $V_G$ is the applied top gate voltage. **c** The bP-PPT can be electrically programmed to 5 states of well-resolved conductance levels using pulses of different voltage amplitudes ($V_{prog}$) with the fixed pulse duration of 20 ms. Note that a negative repressive pulse (−18 V) resets the device to the lowest conductance. Inset: Illustration of the pulse shape of 20 ms in duration and $V_{prog}$ in amplitude. **d** The I–V characteristics of the device at each programmed state, showing the linear conductance ($g_{bP}$). **e** The bP-PPT can be optically programmed using visible pulses to 36 levels in conductance. The inset shows the histogram of the well-separated conductance when the device is programmed repeatedly to three adjacent states indicated with the red dashed ellipse in the histogram. **f** The bP-PPT's photoresponsivity over the near-IR (including the telecom S, C, and L-bands) and the mid-IR ranges. The discontinued spectral region is due to the gap of the laser tunability. The bP-PPT's photoresponsivity can be programmed to two states when its conductance is set to high (state #0 in **e**) or low (state #35 in **e**). The solid lines are guides to the eye.

right edge detection ($\begin{bmatrix} 1 & 1 \\ -1 & -1 \end{bmatrix}$ for top, $\begin{bmatrix} 1 & -1 \\ 1 & -1 \end{bmatrix}$ for left, and $\begin{bmatrix} -1 & -1 \\ 1 & 1 \end{bmatrix}$ for bottom edges)[49–51].

To demonstrate the broadband capability of the bP-PPT array, we encode three different 8-bit grayscale images (Fig. 3b; top: handwritten digits; middle: a husky dog; bottom: a cameraman) using wavelengths in three telecom bands: 1510 nm in the S band, 1550 nm in the C band, and 1590 nm in the L band, respectively. The brightness of each pixel is encoded into the optical power using variable optical attenuators (VOA) and illuminated on the array. Each bP-PPT device is set to have a high responsivity of 60 mA/W to represent 1, or a low responsivity of 20 mA/W to represent −1 (Fig. 2f). The measured photocurrents are normalized and offset to calculate the convolution. The convolved images without any further post-processing are shown in Fig. 3c to f, for right, top, left, bottom edges, respectively. Figure 3g shows the combination of all types of edges, resulting in a clear silhouette of each image. The correlation coefficients between the experimental and the simulated results are over 92% for all three images (Supplementary Fig. 13) Thus, we demonstrate the bP-PPT array's application as an optical frontend capable of multispectral imaging reception and preprocessing.

**bP-PPT CNN for image recognition.** Besides its photoresponsivity, the conductance of the bP-PPT array can also be

programmed to perform MAC operation by measuring the source-drain current $I_{DS} = V_{DS} \cdot g_{bP}$, where $g_{bP}$ is the conductance matrix of the array programmed to represent the weight matrix, $V_{DS}$ is the source-drain voltages applied to the array as the input vector. An optoelectronic convolutional neural network (CNN) thus can be implemented with the array connected to the previous sensory devices, where the optical input image is detected and converted to electrical signals (Fig. 4a, red dashed box). In Fig. 2e, we have demonstrated precise programming of the bP-PPT to 36 discrete levels, ensuring high accuracy in weight training and inference calculation[52,53]. Here, we use the 3 × 3 bP-PPT array to demonstrate a CNN that recognizes images of handwriting numbers "0" and "1" from the MNIST data set. The CNN consists of an input layer that captures a 28 × 28-pixel image, a convolution layer with two 3 × 3 kernels, an average pooling layer followed by an 8 × 2 fully connected (FC) layer (Fig. 4a). The network is trained offline with 12,000 images of the training set with 100 epochs, delivering the final output scores that classify the input image to "0" or "1" with 99% accuracy. The trained network model is remotely programmed into the bP-PPT array by illuminating each pixel with the programming optical pulses. The kernel elements are discretized to accommodate the 36 discrete levels of the programmable states and used consistently in the experiment and simulation (Fig. 4b). For example, the element value 2.00 in kernel 1 (K1), the largest element, is represented by setting a pixel of the bP-PPT to state #35 (in Fig. 2e). By optically programming the 9 pixels of the array to the

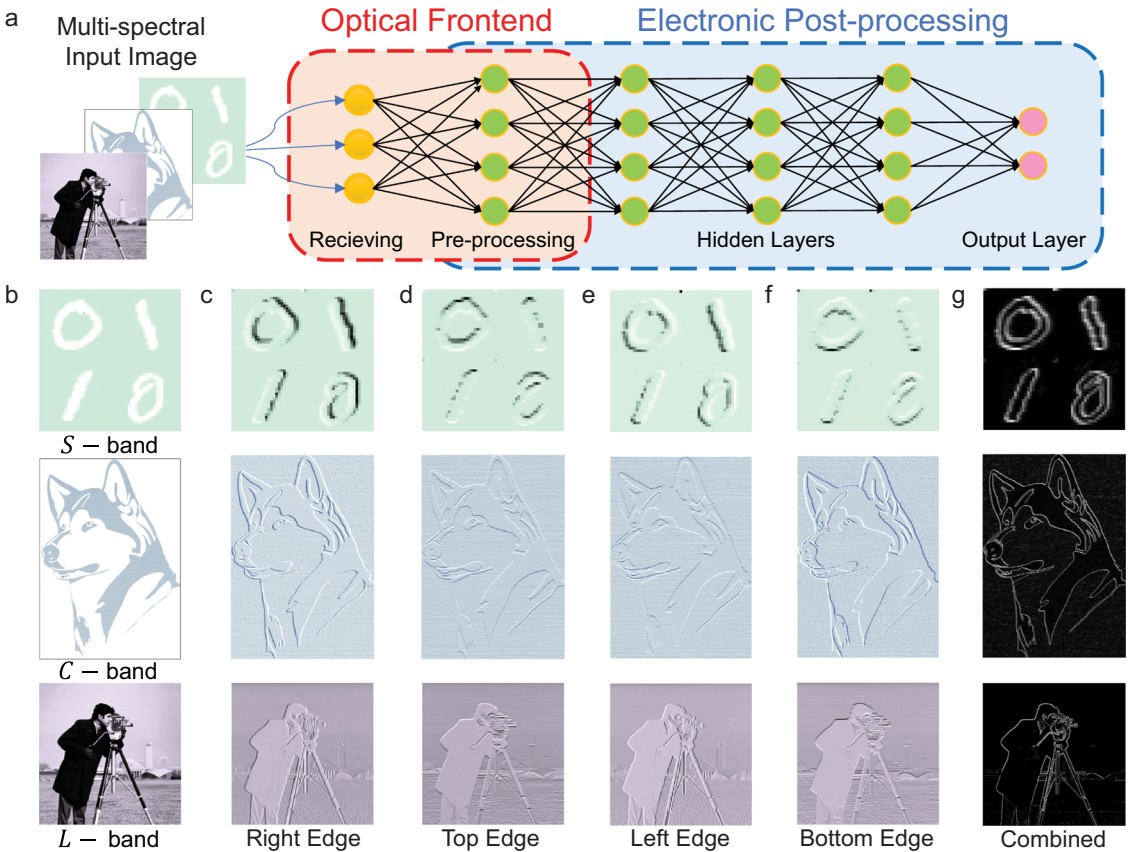

**Fig. 3 bP-PPT array for imaging with in-sensor computing for edge detection. a** The bP-PPT array receives images in multiple wavelength bands. The array's photoresponsivity matrix is programmed to represent the convolution kernel to directly preprocess the images in the optoelectronic domain (red dashed line box). The array's conductance matrix is then programmed to perform inference computation in the electrical domain (blue dashed line box). **b** The original input images encoded in the optical power transmitted in three different telecom bands. Top: handwritten digits (56 × 56 pixels, *S*-band); middle: a husky dog (312 × 222 pixels, *C*-band); bottom: a cameraman (256 × 256 pixels, *L*-band). **c–f** The resultant images after convolution with the right, top, left, and bottom edge kernels, respectively. **g** The final images combining all the edges.

kernel elements, encoding the image pixels in the source-drain voltages, and measuring the source-drain current, the convolution calculation is executed on-chip to obtain the feature maps, followed by the average pooling and FC layers. The two output nodes from the FC layer are activated with the Softmax function and stored as scores to complete the classification task.

To verify the accuracy of the bP-PPT optoelectronic CNN, 100 randomly chosen images of handwritten numbers (48 of "0"s and 52 of "1"s) from the MNIST dataset were tested. The results are compared with the simulated results obtained from a computer. Note that this simulation is different from the first training with 99% accuracy due to the limited 36-level discreteness of the kernel element values. The bar graph in Fig. 4c compares the experimental and the simulated output scores of the two labels "0" and "1" for 50 test cases, which have shown excellent agreement. The gray-tarnished bars in the experimental data highlight the incorrectly recognized cases. The experimental and the simulation results are summarized in the confusion table in Fig. 4d. The bP-PPT array-based CNN reached an accuracy of 92%, comparable with the simulated results (95%).

To summarize, we have demonstrated a phototransistor array based on bP (bP-PPT) that can be programmed electrically and optically by utilizing the stored charges in the gate dielectric stack that has a long retention time. Particularly, our device has a programming precision with a resolution higher than 5-bit, which is among the highest of non-volatile memory devices based on the

charge trapping mechanism[54]. Leveraging its flexible functionality, we use the bP-PPT array to realize vision-sensory functions with in-memory computing. The sensors' programmable photoresponsivity enables in-sensor computing for edge detection on images that are optically encoded and transmitted over a broad infrared band. The same bP-PPT array can also be electrically programmed on the backend to implement a CNN to perform image recognition tasks. Although the demonstrated 5-bit programming precision of our devices is far less precise than that of digital computers, its application in analog in-sensor computing is more suitable for edge computing requiring low power consumption and low latency[37,55–57]. Additionally, the demonstrated programmable photoresponsivity in the near-IR can be extended to a broader range of infrared and further improved by heterogeneous integration of bP with other 2D materials[14,18,58], or optimized for a specific spectral range by varying bP's thickness. It will allow multispectral image processing on edge devices, which can expedite many processes in industrial or biomedical applications[21–29]. Furthermore, recently reported centimeter-scale growth of bP suggests that it is promising to scale the bP-PPT array to an even larger array of megapixels[59]. Thus, the demonstrated multifunctional optoelectronic bP-PPT array, combined with parallel imaging and programming schemes, such as spatial light modulation and wavelength division multiplexing, can realize more complex deep neural networks for machine vision sensors distributed with edge computing.

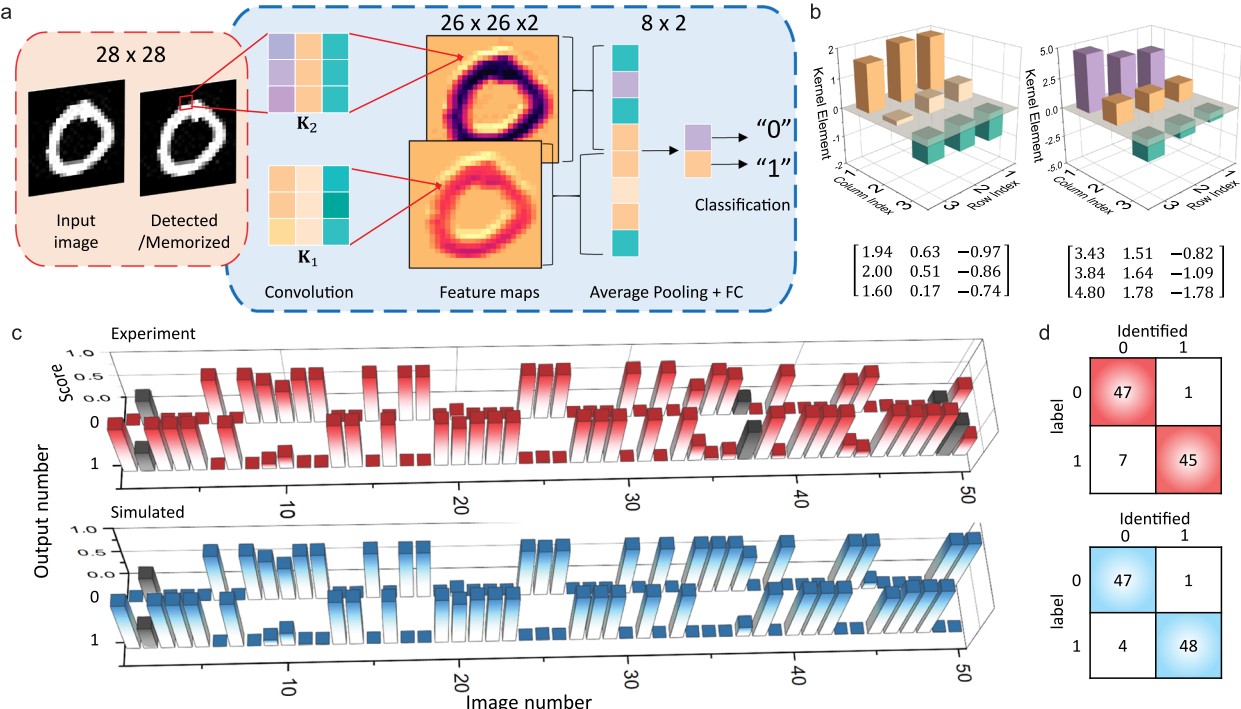

**Fig. 4 bP-PPT array as an optoelectronic CNN for image recognition. a** A CNN model for classifying handwriting numbers "0" and "1" from the MNIST dataset. The CNN consists of two convolution kernels, an average pooling layer, and a fully connected (FC) layer. The images captured by the bP-PPT array are further processed by the bP-PPT array in the electrical domain. **b** The 3 × 3 bP-PPT array is programmed with 5-bit precision to represent two kernels generated by offline training. **c** The experimental results (top, red) for image recognition using the bP-PPT array are compared with the simulation results (bottom, blue). Each bar is the score indicating the possibility of the CNN recognizing an image in the MNIST image library. The incorrectly recognized cases are in gray color. **d** The experimental and simulated confusion table for 100 images from the MNIST dataset. Colored diagonal elements in the table indicate the correctly identified cases.

## Methods

**bP-PPT device fabrication**. Few-layer black phosphorus (bP) with 11 nm thickness was mechanically exfoliated from a bulk crystal (HQ Graphene) and dry transferred using a polydimethylsiloxane stamp onto a silicon substrate with 300 nm-thick thermally grown $SiO_2$. The thickness of the bP flake was identified using an atomic force microscope (Bruker Dimensions Icon). The bP flake with the lateral size of 20 μm × 30 μm was patterned into an array of 4 × 3 pixels with each pixel in the size of 3 μm × 4 μm (Fig. 1c) using electron beam lithography (EBL, JEOL- JBX6300FS) and inductively coupled plasma (ICP) etching based on $SF_6$ chemistry when the ZEP 520 A resist was used as the protective mask. The source and drain contacts made of Ni/Au (5 nm/25 nm) were patterned by steps of EBL, electron beam evaporation, and lift-off in a solvent. The processes of bP exfoliation, thickness measurement, and lift-off of the deposited metal films were all performed in an Ar-filled glovebox with oxygen and water concentration <0.1 ppm to avoid the degradation of the bP flake due to the exposure to moisture and oxygen. Subsequently, the gate dielectric stack of $Al_2O_3$/$HfO_2$/$Al_2O_3$ (AHA) was deposited by atomic layer deposition (ALD) systems. The 6 nm-thick tunneling layer of $Al_2O_3$ was grown on bP at 150 °C by thermal ALD, and the 7 nm-thick storage layer of $HfO_2$ was then grown at 290 °C by plasma-enhanced ALD, followed by the 20 nm-thick blocking layer of $Al_2O_3$. The top gate electrode made of indium-tin-oxide (ITO was patterned by EBL and deposited by a pulsed sputtering system (Evatec LLS EVO).

**Measurement setup**. The bP-PPT devices were wire-bonded to the 64-pin chip holder as shown in Supplementary Fig. 14. For electrical measurements, each device's gate, source, and drain contacts were connected to 3 pins of the holder. Since we have 12 bP-PPT devices, we can simultaneously measure the conductance or photocurrent of several devices using a set of source-measurement unit (SMU) modules. For the optical input, both 780 nm and tunable telecom-band lasers were aligned and focused onto each device. We programmed the devices using a 780 nm laser diode (LP785SF20, Thorlabs) with tunable output power and pulse width. The optical images were input to the devices by modulating the intensity of a tunable telecom band laser (TSL-210, Santec Corporation) using a variable optical attenuator (EVOA1550A, Thorlabs). Optical programming of the bP-PPT array was achieved by using a spatial light modulator. A data acquisition system was used to measure the device array and the output was feedback to the device array to realize a network. To characterize the devices' photoresponsivity in the broadband infrared spectrum, a tunable infrared laser (Firefly, M Squared) was used with its

signal output in the wavelength range of 1.5–1.8 μm and its idler output in the wavelength range of 2.6–3.1 μm. A $CaF_2$ lens was used to focus the infrared light onto the bP-PPT devices.

## Data availability

The data that support the findings of this study are available from the corresponding author upon reasonable request.

## Code availability

No custom computer code or mathematical algorithm is used to generate the results that are reported in this study.

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

## Acknowledgements

We acknowledge the funding support provided by the ONR MURI (Award No. N00014-17-1-2661) and the National Science Foundation (NSF MRSEC DMR-1719797). Part of this work was conducted at the Washington Nanofabrication Facility/Molecular Analysis Facility, a National Nanotechnology Coordinated Infrastructure (NNCI) site at the University of Washington with partial support from the National Science Foundation via awards NNCI-1542101 and NNCI-2025489.

## Author contributions

S.L., R.P., and M.L. conceived the research. S.L. fabricated the devices, performed the measurements. S.L. R.P., and C.W. analyzed the data. S.L, R.P., and M.L. co-wrote the manuscript with contributions from all authors.

## Competing interests

The authors declare no competing interests.
