## [Peer Review File · Nature Communications]

Programmable black phosphorus image sensor for broadband optoelectronic edge computing

Programmable black phosphorus image sensor for broadband optoelectronic edge computingREVIEWER COMMENTS

Reviewer #1 (Remarks to the Author):

This is really a nicely written and robustly performed work, with the novelty arising from using the black phosphorus (bp) photo transistors for in-sensor, in-memory computing (sensing, memory, and computing take place all in the front-end layer). The in-sensor, in-memory computing is a timely subject, and the authors bring the excitement by performing the work with bp photo transistor that has the response in the IR spectral range, which is relevant to a lot of edge applications. I think this work will be of broad interest, and would like to recommend the publication of this manuscript, if the authors can address the following points.

1. While 2D materials and bp are indeed very interesting, and are particularly so because of their potential for vdW stacking to build a variety of heterostructure with varying bandgaps for rich spectral responses, they are technologically still not fully mature, for example, as compared to CMOS imager technology used for both image sensing and processing with a very large number of devices integrated with a high yield. The authors do NOT have to mention all of this, but at the least, they need to provide some balance in the manuscript by refraining from using superlatives. For example, I suggest that the authors do NOT use (so please tone down) such phrases as "...are uniquely advantageous in realizing such intelligent visionary sensors..." in abstract and "...tremendous potential in optoelectronics...". This is to keep the scholarly balance of the manuscript. Throughout the manuscript, the authors can tone down such exaggerated statements. Such will not compromise the quality of the work they have performed and I believe it will be still appealing to the broad science and technology community.

2. Have the authors performed the work with various wavelengths? Or the wavelength tunability can be done by adjusting the number of bP layers stacked? The authors do NOT have to perform any new experimental work, but if they could provide a perspective for the spectral richness (they do discuss it in the general context, but can they do it based on their experimental results or from the point of view of their specific device), that would further strengthen the paper and broaden its perspective. For at the end of the day, what can be uniquely interesting with the 2D materials and bPs would be their ability to stack and the resulting richness in their physical properties.

3. Another scholarly balance issue - while a lot of people perform the analogue MAC operation (as the authors do in the manuscript) by arranging memories in an array and performing multiplication via Ohm's law and performing accumulation via current summation, and while this is certainly a very interesting approach which is also very powerful from the power saving point of view (much less power than the digital MAC operation), such analogue/physical computing trades the power saving with the computing accuracy (the analogue computing accuracy cannot compete with the digital computing with so many (e.g., 32-b) floating points. But for edge sensing, where the power saving is absolutely important and the lowered accuracy can be afforded, still the analogue MAC operation can have very (very) powerful applications. The authors should describe this context so that the drawback is not concealed but the adequate application is emphasized. The recent article, Nature Electronics 4, 635-644 (2021), "Neuromorphic electronics based on copying and pasting the brain", by Ham et al articulates this big picture on analogue in-memory MAC computing, including its suitability in edge computing, in its review of the contemporary neuromorphic engineering, so I suggest that the authors cite this paper properly in their description of what I have suggested in the foregoing.

4. If the barrier is lower for charge storage, it might be easier to write but also its retention time might be compromised (that is a sort of general belief in the community). Does this apply to the bP device described here? Can the authors address this point briefly in the paper?

5. The physical array size is not large. This is a research device, so it is totally fine, I think, but for the future technology driving in a longer term, do the authors envision a possibility for a larger array size? Such would require a large area bP, so flakes might not be sufficient. Do people try to grow a large-area bP via chemical deposition? (e.g., MOCVD is done for MoS₂ for large area growth). A perspective on this can be briefly discussed in the concluding paragraph to further

broaden the perspective of the work.

Once again, I believe this is an excellent work with very appealing results, and the incorporation of the points I mentioned above, I believe, will further strengthen the manuscript.

Reviewer #2 (Remarks to the Author):

In this paper the authors report a black phosphorus based infrared image sensor. Although the paper is well structured, this reviewer is not convinced by the novelty the work represents. Black phosphorus has been used in image enhancement and pattern recognition by deploying similar methods for different wavelengths. The paper does not even refer to them and benchmark this against existing similar studies.

Infact such operations have been demonstrated with black phosphorus and a range of other materials without even requiring a complex device structure and in some cases even 2 terminal configurations.

As such, this work does not provide an advancement in knowledge for this journal.

Before considering the submission elsewhere, there are several suggestions I would like to make to improve the paper:

1. Clearly benchmark the work against existing similar or more advanced works
2. The optical image of the exfoliated flake clearly shows regions of varying thickness. How did the authors ensure each pixel is of the same thickness?
Also, why 11 nm was chosen as the key thickness value for this study.
3. There is hardly any characterisation data provided for the material. How much of it is oxidised.
4. Why did the authors focus only on the IR range when the black phosphorus can be even more broadband into the visible at the thickness used here
5. There is no information provided on how the read out from the multiple electrode pairs was achieved in this case. This is important information for the readers.
6. There are many studies that report that ICP etching tends to result in defects in the black phosphorus crystals. Did the authors characterise the material before and after etching to ensure properties do not change. And if they do what changes. This missing analysis is crucial.
7. Finally no lifetime data is provided. How long do the devices keep working in a stable manner

In conclusion, in its present form there are several flaws and missing information that needs to be addressed. The paper does not report a significant enough advance to warrant publication in Nature Communications however the above points should be well considered in my opinion before publication in any journal.

Reviewer #3 (Remarks to the Author):

In this work, Seokhyeong Lee et al. demonstrates the multifunctional image sensor that combining the functions of multispectral imaging and analog in-memory computing based on the BP programmable phototransistors array. Based on the charge trapping in the AHA dielectrics through electrical or optical method, the conductance and photoresponsivity of the BP phototransistor can be precisely programmed to realize the in-sensor CNN. As the sensor, BP phototransistors array can not only receive optical images which are optically encoded and transmitted over a broad spectral in the infrared range, but also electrically perform inference computation to process and recognize the images. As a result, the demonstrated multifunctional optoelectronic BP phototransistors array holds the promise to realize more complex deep neural networks for machine vision sensors distributed with edge computing. The paper is well organized, and the main results are convincing and interesting. Furthermore, I suggest that authors revise the manuscript to address the issues discussed below.

1. The working principle of the BP floating gate device under electrically or optically program processes should be demonstrated more detailly. In page 3, authors state "charges (electrons or holes) can tunnel from the top gate through the thin Al₂O₃ barrier layer to be stored in the HfO₂ layer", which doesn't seem to be consistent with the information in Figure 2a.
2. In page 3, author demonstrate the advantages of the AHA gate dielectric such as reliable and faster operation due to the utilized HfO₂ dielectric layer with a lower formed barrier height. How fast can this device work? The charge trapping process and retention performance rely on the barrier between HfO₂ and Al₂O₃, the barrier between Al₂O₃ and BP, as well as the thickness of the tunneling layer. Please give the comprehensive analysis.
3. What's the influence of the AHA thickness on the electrical and optical performance of the BP floating gate device?
4. How to evaluate the degradation problem of this BP floating gate device? Because this hinders the practical application of the device.
5. For Figure 2c, it just shows the retention time about 2000s, which is not consistent with the description in the page 4.
6. Authors demonstrate that the device can realize 3 bits or even 5 bits conductance states. How to differentiate the different states? Is there some criterion?
7. In Figure 3a, authors demonstrate that the array can act as both the optical frontend to receive and preprocess optical images and an electrical processor with in-memory computing to post-process the images. However, the connection between two applications (edge detection and image recognition) is not demonstrated clearly. So authors can extend the descriptions of the connections between two array applications more to help reader understand its significance.
8. The edge detection in Figure 3 is mainly based on the 2×2 bP-PPT array. How to evaluate the detection results?

The Authors' Response to Reviewers' Comments

Reviewer #1:

This is really a nicely written and robustly performed work, with the novelty arising from using the black phosphorus (bp) photo transistors for in-sensor, in-memory computing (sensing, memory, and computing take place all in the front-end layer). The in-sensor, in-memory computing is a timely subject, and the authors bring the excitement by performing the work with bp photo transistor that has the response in the IR spectral range, which is relevant to a lot of edge applications. I think this work will be of broad interest, and would like to recommend the publication of this manuscript, if the authors can address the following points:

Our response: We thank the reviewer for the very positive and insightful comments on our work.

1. While 2D materials and bp are indeed very interesting, and are particularly so because of their potential for vdW stacking to build a variety of heterostructure with varying bandgaps for rich spectral responses, they are technologically still not fully mature, for example, as compared to CMOS imager technology used for both image sensing and processing with a very large number of devices integrated with a high yield. The authors do NOT have to mention all of this, but at the least, they need to provide some balance in the manuscript by refraining from using superlatives. For example, I suggest that the authors do NOT use (so please tone down) such phrases as "...are uniquely advantageous in realizing such intelligent visionary sensors..." in abstract and "...tremendous potential in optoelectronics...". This is to keep the scholarly balance of the manuscript. Throughout the manuscript, the authors can tone down such exaggerated statements. Such will not compromise the quality of the work they have performed and I believe it will be still appealing to the broad science and technology community.

Our response: We thank the reviewer very much for the suggestions to improve the soundness and the scholarly balance of our paper. We have revised the manuscript accordingly to tone down the statements about 2D materials.

- 2. Have the authors performed the work with various wavelengths? Or the wavelength tunability can be done by adjusting the number of bp layers stacked? The authors do NOT have to perform any new experimental work, but if they could provide a perspective for the spectral richness (they do discuss it in the general context, but can they do it based on their experimental results or from the point of view of their specific device), that would further strengthen the paper and broaden its perspective. For at the end of the day, what can be uniquely interesting with the 2D materials and bPs would be their ability to stack and the resulting richness in their physical properties.

Our response: Because of bp's narrow bandgap, the bp phototransistor (bp-PPT) we demonstrated indeed can work in a wide range of wavelengths from the near-IR (the telecom S-C-L bands) to mid-infrared. This is shown in Figure 2f, where the device's photoresponsivity is measured over a broad infrared band using our tunable laser. Notably, the responsivity is programmable between two states, which we utilized to realize the in-sensor computing as demonstrated in Figure 3. Indeed, one of the uniquely interesting aspects of 2D materials is their tunability. For bp, its bandgap is tunable by the number of layers, which can be optimized for photoresponsivity in a specific wavelength range. Heterogeneous integration of multiple 2D materials in a stack can further tune and optimize the optoelectronic functions. Based on the reviewer's suggestion, we have added the following discussions on this perspective in the revised manuscript.

Additionally, the demonstrated programmable photoresponsivity in the near-IR can be extended to a broader range of infrared and further improved by heterogeneous integration of bP with other 2D materials^{14,18,20}, or optimized for a specific spectral range by varying bP's thickness.

- 3. Another scholarly balance issue - while a lot of people perform the analogue MAC operation (as the authors do in the manuscript) by arranging memories in an array and performing multiplication via Ohm's law and performing accumulation via current summation, and while this is certainly a very interesting approach which is also very powerful from the power saving point of view (much less power than the digital MAC operation), such analogue/physical computing trades the power saving with the computing accuracy (the analogue computing accuracy cannot compete with the digital computing with so many (e.g., 32-b) floating points. But for edge sensing, where the power saving is absolutely important and the lowered accuracy can be afforded, still the analogue MAC operation can have very (very) powerful applications. The authors should describe this context so that the drawback is not concealed but the adequate application is emphasized. The recent article, *Nature Electronics* 4, 635-644 (2021), "Neuromorphic electronics based on copying and pasting the brain", by Ham et al articulates this big picture on analogue in-memory MAC computing, including its suitability in edge computing, in its review of the contemporary neuromorphic engineering, so I suggest that the authors cite this paper properly in their description of what I have suggested in the foregoing.

Our response: We thank the reviewer for the very insightful comments. Following the reviewer's suggestions, we have cited Ham et. al. and added the following discussion on page 8 about analog computing's limitations and its unique advantages in edge computing.

Although the demonstrated 5-bit programming precision of our devices is far less precise than that of digital computers, its application in analog in-sensor computing is more suitable for edge computing requiring low power consumption and low latency [Ref. 58: *Nat Elec* 4, 635-644 (2021)].

- 4. If the barrier is lower for charge storage, it might be easier to write but also its retention time might be compromised (that is a sort of general belief in the community). Does this apply to the bP device described here? Can the authors address this point briefly in the paper?

Our response: Indeed, for devices using trapped charges, there is a trade-off between the retention time and the barrier height, which also applies to our bP device. To address this point, we have performed a theoretical analysis using the Fowler-Nordheim tunneling current model for the triangle barrier of bP/AHA/ITO in our device to estimate the retention time and the electrical and optical programming speed. Please see our response to reviewer #3 below. We have included this analysis in the revised S.I. Note 4,5 and 6.

- 5. The physical array size is not large. This is a research device, so it is totally fine, I think, but for the future technology driving in a longer term, do the authors envision a possibility for a larger array size? Such would require a large area bP, so flakes might not be sufficient. Do people try to grow a large-area bP via chemical deposition? (e.g., MOCVD is done for MoS₂ for large area growth). A perspective on this can be briefly discussed in the concluding paragraph to further broaden the perspective of the work?

Our response: We thank the reviewer for bringing up the important issue of growing large area bP. Indeed, large-area growth of bP was recently reported by Wu, Z. *et al.* ("Large-scale growth of few-layer two-dimensional black phosphorus," *Nat. Mater.* **20**, 1203–1209 (2021)), in which they demonstrated

centimeter-scale bP growth on a mica substrate using pulsed laser deposition. This is very good news for bP research as its large area growth has been lacking for many years. Following the reviewer's suggestion, we added the following discussion on the perspective of large-scale integration in the concluding paragraph.

Furthermore, recently reported centimeter-scale growth of bP suggests that it is promising to scale the bP-PPT array to an even larger array of megapixels⁵⁹. Thus, the demonstrated multifunctional optoelectronic bP-PPT array, combined with parallel imaging and programming schemes, such as spatial light modulation and wavelength division multiplexing, can realize more complex deep neural networks for machine vision sensors distributed with edge computing.

Response to Reviewer #2:

In this paper the authors report a black phosphorus based infrared image sensor. Although the paper is well structured, this reviewer is not convinced by the novelty the work represents.

Black phosphorus has been used in image enhancement and pattern recognition by deploying similar methods for different wavelengths. The paper does not even refer to them and benchmark this against existing similar studies.

In fact such operations have been demonstrated with black phosphorus and a range of other materials without even requiring a complex device structure and in some cases even 2 terminal configurations.

As such, this work does not provide an advancement in knowledge for this journal.

Before considering the submission elsewhere, there are several suggestions I would like to make to improve the paper:

Our response: We thank the reviewer for the insightful comments. However, we respectfully disagree with respect to the novelty of our work. We have cited a few references (Ref. 8, 9) on network image sensors using transition metal dichalcogenides (TMD):

- i) Ref. 8: Jang, H., *et al.* An Atomically Thin Optoelectronic Machine Vision Processor, *Adv. Mater.* **32**, 36 (2020)
- ii) Ref. 9: Mennel, *et al.*, Ultrafast machine vision with 2D material neural network image sensors,” *Nature* **579**, 62–66 (2020))

We recently noticed a related work using bP:

- iii) Ahmed, T. *et al.*, Fully Light-Controlled Memory and Neuromorphic Computation in Layered Black Phosphorus. *Adv. Mater.* **33**, 2004207 (2021)

And while our manuscript was under review, another related paper was published, which we were completely unaware of:

- iv) Zhang, Z., Wang, S., Liu, C. *et al.* All-in-one two-dimensional retinomorphic hardware device for motion detection and recognition. *Nat. Nanotechnol.* (2021).
<https://doi.org/10.1038/s41565-021-01003-1>

Compared with these prior arts, our work has the following significant differences and novelties:

1. We demonstrated non-volatile programming of the devices' photoresponsivity (Figure 2f) over a broad optical spectral range (1.5-3.1 μm) and utilized it to perform both direct in-sensor processing of received optical signals (edge detection in Figure 3). In contrast, the references above only demonstrate programming the 2D devices' electrical conductance by optical illumination. None of them demonstrated direct in-sensor imaging processing.

For example, Jang, H. *et al.* demonstrated an optoelectronic machine vision processor based on the optical programming of the persistent photoconductivity (PPC), which is different from photoresponsivity. In their device, the photo-excited charges are trapped in the defects and impurity sites in the channel of the MoS₂ FET, modulating the electrical conductance. Ahmed, T. *et al.* used a similar PPC effect based on bP transistors to realize a neural network that the trapped charges in the PO_x are programmed and erased by UV light. Zhang, Z. *et al.* program the PPC states of the bP by optically controlling the density of charges trapped in the WSe₂ layer. None of the above works achieved nonvolatile control of the photoresponsivity and used the devices to simultaneously receive and process input optical images. Those are the major differences from our work.

- We also demonstrated programming of the persistent photoconductivity as one of four operation modes in our bP devices (Figure 2e) which has shown very high precision with multiple programmable states (36 levels) and a retention time of hours. Our device set the record of the largest number of states (36 levels) for the demonstrated charge trap transistors or floating gate transistors based on 2D materials (see Table S1 below).
- We engineered the multilayer AHA structure with the HfO₂ trapping layer to store electrically injected charges with higher density and longer retention time than the native oxide, material defects, or impurity traps as used in the previous studies.

Therefore, we believe our work presents significant novelty and advances from prior works. Together with our detailed response to all the reviewers' comments below and a major revision of our manuscript, we hope that we have adequately addressed the concerns and convinced the reviewers about our work's novelty.

1. Clearly benchmark the work against existing similar or more advanced works

Our response: We thank the reviewer for suggesting benchmark against the prior works, which is done in Table S1 below. The distinctions and novelties of our work have been discussed above. We have also revised our manuscript to further clarify our work's differences and included the following table in the S.I.

Table S1. Comparison of key features and performance of our device with prior works.

Ref	Material		Programming method	Physical Attribute		CNN Accuracy	# of stable states
	Channel	Storage		Weight	Input		
This work	bP	AHA	Electrical & Visible light	g, R	E, O	92	36
1	bP	PO _x	UV light	g	E	90	N.D.
2	WSe ₂	h-BN	Electrical	g, R	E, O	90	N.D.
3	MoS ₂	AlO _x	Electrical & Visible light	g	E	94	4
4	bP	AHA	Electrical	g	E	N.D.	2
5	bP	cPVP	Electrical	g	E	N.D.	5
6	MoS ₂	graphene	Electrical	g	E	N.D.	N.D.
7	InSe	graphene	Electrical	g	E	N.D.	16
8	WSe ₂	No memory	Electrical	R	O	99	N.D.

* Input E: electrical, O: optical

* Weight: g for conductance, R for responsivity

* *N.D.*: Not demonstrated

* AHA: Al₂O₃/HfO₂/ Al₂O₃ stack

2. The optical image of the exfoliated flake clearly shows regions of varying thickness. How did the authors ensure each pixel is of the same thickness?

Also, why 11 nm was chosen as the key thickness value for this study.

Our response: We thank the reviewer for raising this question. As shown in revised Fig. S9 (also included below), the mechanically exfoliated bP has several regions with different thicknesses. We fabricated all the pixels within a large region with uniform optical contrast, as outlined by the red dotted line, indicating uniform thickness. The thickness of the region is also confirmed with atomic force microscopy.

There are several reasons we choose 11 nm as the thickness of bP in our devices:

1. To achieve broadband infrared optical response, the thickness-dependent bandgap of bP needs to be more than 5 nm (Ref. Li, L., et al. “Direct observation of the layer-dependent electronic structure in phosphorene.” *Nature Nanotech* **12**, 21–25 (2017).)
2. On the other hand, to reach effective electrostatic doping, the bP flake needs to be thinner than 20 nm (Ref. “Black phosphorus field-effect transistors,” *Nature Nanotech* **9**, 372–377 (2014)).
3. bP thinner than ~10 nm suffers lowered carrier mobilities because thinner flakes are more susceptible to charge impurities at the interface that are otherwise screened by the induced charge in thicker flakes.

Therefore, a thickness of ~11 nm is optimal for the device demonstrated in our work.

3. There is hardly any characterisation data provided for the material. How much of it is oxidised.

Our response: Following the reviewer’s suggestion, we have included more material characterization results in the revised S.I. As for the oxidation of bP, it is indeed a concern. To mitigate that, we exfoliated and transferred bP in an Ar-filled glovebox. The device was immediately loaded into the ALD chamber to deposit the AHA multilayers, which encapsulate the bP flake to prevent oxidation and degradation. This has been a practice reported in the literature, which generally leads to oxidation of only a few layers (Pei, J., *et al.* “Producing air-stable monolayers of phosphorene and their defect engineering.” *Nat Commun* **7**, 10450 (2016); Deng, B., *et al.* “Efficient electrical control of thin-film black phosphorus bandgap.” *Nat Commun* **8**, 14474 (2017)). Thanks to the 35 nm thick AHA encapsulation layer, our device shows long-term stability with persistent electrical and optical properties for more than 3 months after fabrication. We discuss that in the response to question 7. Furthermore, Raman spectroscopy shows no sign of P_xO_y or H_xPO_y forming during the fabrication process (please see our response to question 6). Thus, we expect that

the oxidation of bP flake is no more than 3 layers (or 1.5 nm), which, if any, marginally affects the optical and electrical properties of the bP-PPT device.

4. Why did the authors focus only on the IR range when the black phosphorus can be even more broadband into the visible at the thickness used here.

Our response: The photoresponsivity of black phosphorus certainly extends to the visible band. We have utilized both the visible and the IR range by engineering the device with the AHA stack: IR light to input images for optoelectronic in-sensor computing (Figure 3); visible light to optically program the device by activating the trapped charges (Figure 2b and e) and process the encoded images such as pattern recognition (Figure 4).

5. There is no information provided on how the read out from the multiple electrode pairs was achieved in this case. This is important information for the readers.

Our response: Following the reviewer's suggestion, we have included a detailed description and discussion about our measurement scheme in the revised S.I. The bP-PPT devices were wire-bonded to a 64-pin chip holder as shown in Fig. S14 (also shown below). As for the electrical measurements of the bP devices, each device was wired to 3 pins for connections to the gate, source, and drain. Since we have 12 bP-PPT devices, we can simultaneously measure the conductance or photocurrent of several devices using a set of SMU modules. For the optical input, both 780 nm and telecom-band laser beams were aligned and focused onto one device. We programmed the devices using a 780 nm laser diode (LP785SF20, Thorlabs) with tunable output power and pulse width. The optical images were input to the devices by modulating the intensity of the telecom laser (TSL-210, Santec Corporation) using a variable optical attenuator (EVOA1550A, Thorlabs), and detected by measuring the photocurrents of the bP-PPT devices. LD: Laser Diode, DM: Dichroic Mirror.

Supplementary Figure 14. Measurement scheme for the bP-PPT array.

6. There are many studies that report that ICP etching tends to result in defects in the black phosphorus crystals. Did the authors characterise the material before and after etching to ensure properties do not change. And if they do what changes. This missing analysis is crucial.

Our response: We thank the reviewer for raising this concern about the bP degradation. Following the suggestion, we have included new characterization results of bP in the S.I. Note 3.

First, during the ICP etching process, the bP channel region of each bP-PTT device was protected with 520-nm thick resist (ZEP 520 A), which is thick enough to prevent any exposure of the bP to the plasma and degradation that may cause. The resist was then removed with solvent (NMP, N-Methyl-2-pyrrolidone) in an Ar-filled glovebox. NMP has no detrimental effects on bP flake as suggested by Kang, J., *et al.* (“Solvent exfoliation of electronic-grade, two-dimensional black phosphorus,” *ACS nano* **9.4**, 3596-3604(2015)).

Second, to characterize the impact of the process, we repeated the ICP etching process using a freshly exfoliated bP flake and performed AFM and Raman spectroscopy measurements before and after the process. The results are included in the revised S.I. and shown in the figure below. As shown in panel **a** of the figure, the freshly exfoliated bP flake has a flat, clean surface with a thickness of 17 nm. We then use e-beam lithography and ICP etching to pattern the bP to two small rectangles. After the patterning process, the bP flake has the same thickness (17 nm) and its surface remains to be flat and clean as the just exfoliated flake, indicating the bP is well protected by the resist during the process without observable degradation. As discussed in Peng, L, *et al.* (“Black Phosphorus: Degradation Mechanism, Passivation Method, and Application for In Situ Tissue Regeneration.” *Advanced Materials Interfaces* **7**, 23 (2020)), the degradation will increase the surface roughness of the bP flake, which can be visualized in the optical and AFM images.

Lastly, as reported by Naqvi, B.A., *et al.* (“Visualizing Degradation of Black Phosphorus Using Liquid Crystals.” *Sci. Rep.* **8**, 12966 (2018)), the degradation of bP is attributed to the formation of P_xO_y or H_xPO_y , which has Raman peaks in the range of 800 to 1000 cm^{-1} . We measured Raman spectroscopy of the bP flake after the patterning processes (e-beam lithography, ICP etch, resist removal). The spectrum (panel **b**) shows no signs of P_xO_y or H_xPO_y , thus confirming that oxidation and degradation of bP during the fabrication process are negligible.

7. Finally no lifetime data is provided. How long do the devices keep working in a stable manner.

Our response: We agree with the reviewer that more lifetime measurement data is important to ensure the stability of the device's operation. We have evaluated the performance of the bP-PPT devices over a long period of time (up to 3 months) and include the results in the revised S.I. As shown in the figure below (also in Figure S11), the gate modulation, memory window, on-off ratio, and retention time of our devices remain consistent even 3 months after they were fabricated.

The excellent stability and longevity of the devices can be attributed to the protection by the AHA stack and our meticulous fabrication process that minimizes degradation and contamination of the bP. The 35 nm AHA ($Al_2O_3/HfO_2/Al_2O_3$) layers are deposited by atomic layer deposition (ALD), which effectively passivate and protect the bP. Our results are consistent with the literature (Gamage, S., *et al.* ("Nanoscopy of black phosphorus degradation." *Advanced Materials Interfaces* 3, 12, 1600121 (2016)), which reported that 20 nm ALD alumina (Al_2O_3) can conformally coat the device and protect the thin bP flake from degradation for more than 90 days.

To further confirm the devices' stability during the operations, we also conducted the endurance test that showed multi-state programming with excellent repeatability over 200 cycles. The result is shown in the figure below and in Figure S1c.

Supplementary Figure 1. (a) Time trace of state retention for two conductive states. (b) 3 bits (8 states) programming with electrical pulses. The first 18 V voltage pulse sets the device to the highest conductance state. Following depressive pulses of 20-ms with different voltage from -4 V to -12 V set the device to 8 different states (color coded). (c) Endurance test of the bP-PPT devices for 200 cycles of repeated procedure described in (b).

Response to Reviewer #3:

In this work, Seokhyeong Lee et al. demonstrates the multifunctional image sensor that combining the functions of multispectral imaging and analog in-memory computing based on the BP programmable phototransistors array. Based on the charge trapping in the AHA dielectrics through electrical or optical method, the conductance and photoresponsivity of the BP phototransistor can be precisely programmed to realize the in-sensor CNN. As the sensor, BP phototransistors array can not only receive optical images which are optically encoded and transmitted over a broad spectral in the infrared range, but also electrically perform inference computation to process and recognize the images. As a result, the demonstrated multifunctional optoelectronic BP phototransistors array holds the promise to realize more complex deep neural networks for machine vision sensors distributed with edge computing. The paper is well organized, and the main results are convincing and interesting. Furthermore, I suggest that authors revise the manuscript to address the issues discussed below.

Our response: We thank the reviewer for the positive and insightful comments on our work.

1. The working principle of the BP floating gate device under electrically or optically program processes should be demonstrated more detailly. In page 3, authors state “charges (electrons or holes) can tunnel from the top gate through the thin Al₂O₃ barrier layer to be stored in the HfO₂ layer”, which doesn’t seem to be consistent with the information in Figure 2a.

Our response: We thank the reviewer for finding this error in our manuscript. We have corrected it in the revised manuscript as:

the charges tunnel from the bP channel through the thin Al₂O₃ barrier layer to be stored in the HfO₂ layer.

Also, we improved the description of the working principle in the revised manuscript and the S.I. by including a theoretical model based on the Fowler-Nordheim tunneling to analyze the electrical and optical programming process of the bP-PPT devices. It is also included below:

Supplementary Note 4. The working principle of the bP-PPT device can be modeled by the Fowler-Nordheim tunneling (FN tunneling), which explains the charge trapping and de-trapping mechanism by electrical and optical control.

Fig. S6a depicts the band alignment of the bP channel, Al₂O₃ tunnel layer, and HfO₂ charge storage layer. When a sufficiently large voltage is applied to the top ITO gate (Fig. 1e in the main text), the large electric field across the tunneling layer can lead to the FN tunneling. A tunneling current will be injected into the HfO₂ trapping layer, where charges are trapped at trapping sites with energy in the bandgap (Ref. 12: Gritsenko, V. A., *et al.* “Electronic properties of hafnium oxide: A contribution from defects and traps”. *Physics Reports*, 613, 1-20. (2016)). After the applied gate voltage is removed, these metastable trapped charges remain in the HfO₂ layer and induce effective gating to the bP channel, modulating its optical and electric properties. Fig. S6b illustrates the band diagram under the built-in electric field by these trapped charges.

Under optical illumination, the trapped electrons can be excited and escape from the trap site to tunnel back to the bP-channel, facilitated by the built-in field as in Fig. S6b. The amount of tunneling charges can be precisely controlled by the optical power and pulse duration to realize 36 intermediate states. The electric field distributions with and without gate voltage are depicted in Fig. S6c and d, for different charge densities in the charge storage layer (long-dashed line, short-dashed, and straight lines for the highest, moderate, and no charges, respectively). To estimate the electrical programming/erasing speed and optical programming speed, we calculate the FN tunneling current with a triangle barrier of bP/AHA/ITO

device. The quantum tunneling transmission function $TC(\xi)$ is first calculated using the Wentzel-Kramers-Brillouin (WKB) approximation, and is given by (ref. 13: *Journal of Applied Physics* 105, 094103 (2009)):

$$TC(\xi) = \exp\left(-\frac{2}{\hbar} \int_0^{t_{ox}} \sqrt{2m_{ox}(V(x) - \xi)} dx\right) \quad (1)$$

where t_{ox} is the thickness of the tunneling layer (Al_2O_3), m_{ox} is the effective mass of the carriers in the tunneling layer, $V(x)$ is the potential function of the triangle-barrier, and ξ is the energy of incident carriers referenced to the Fermi energy of the electrode.

When the gate voltage is applied, the electric field across the tunneling Al_2O_3 layer is calculated using the capacitor model:

$$V_G = \frac{\sigma_1}{\epsilon_{AlO}} t_{blc} + \frac{\sigma_1}{\epsilon_{HfO_2}} (t'_{str}) + \frac{\sigma_1 + \sigma_{HfO_2}}{\epsilon_{HfO}} (t_{HfO} - t'_{str}) + \frac{\sigma_1 + \sigma_{HfO_2}}{\epsilon_{AlO}} t_{tnl} \quad (2)$$

$$E_{ox} = \frac{V_{tnl}}{t_{tnl}} = \frac{\sigma_1 + \sigma_{HfO_2}}{\epsilon_{AlO}} \quad (3)$$

where σ_1 , σ_{HfO_2} are the charge density at the interface between the gate electrode and the blocking layer, and the charge density stored in the HfO_2 layer, ϵ_{AlO} , ϵ_{HfO} are the dielectric constant for Al_2O_3 and HfO_2 , t_{tnl} , t_{HfO} , t_{blc} and t'_{str} are the thickness of tunneling layer, charge storage layer, blocking layer, and the position for the barycenter of the stored charge, respectively. We assume the charge density is uniformly distributed in the HfO_2 layer so that the barycenter is assumed as the center of HfO_2 . Accordingly, the tunneling current can be estimated as (ref. 13: *Journal of Applied Physics* 105, 094103 (2009)):

$$J_{FN} = \frac{m_e q^3}{8\pi m_{ox} h q \Phi_B} E_{ox}^2 \exp\left(-\frac{4\sqrt{2m_{ox}}}{3\hbar q E_{ox}} (q\Phi_B)^{\frac{3}{2}}\right) \quad (4)$$

where q , h , m_e , and Φ_B are electron charge, Planck's constant, electron mass, potential energy barrier at the bP and Al_2O_3 interface, respectively.

The stored charge density in the HfO_2 layer ($\sigma_{HfO_2}(t)$) (negative for electrons and positive for holes) is the integration of the tunneling current density over the operation time t ,

$$\sigma_{HfO_2}(t) = \int_0^t J_{FN}(t') dt' \quad (5)$$

The stored charge density after the programming by the gate pulse can induce effective gating to the bP channel and modulate its conductance. Meanwhile, σ_{HfO_2} induces built-in electric field across the tunneling layer and causes a leakage current to the bP channel. The tunneling current of the leakage process depends on the tunneling coefficient and the carrier density regarding the process, which can be described by the Tsu-Esaki formula [ref. 14: *Appl. Phys. Lett.* 22, 562 (1973)]:

$$J_{leak} = -q \frac{dn_{stored}}{dt} = \frac{q}{4\pi^3 \hbar} \int_{\xi_{min}}^{\xi_{max}} TC(\xi) g(\xi) f(\xi) d\xi \quad (6)$$

where $g(\xi)$ and $f(\xi)$ are the density of state of charge carriers and distribution function, respectively. For simplicity, we assume charge trapping in HfO_2 is dominated by one type of traps so simplify $g(\xi)$ with delta function, $g(\xi) = n_{stored} \delta(\xi - \xi_{trap})$. $f(\xi)$ is Fermi-Dirac distribution function. Then, since the tunneling coefficient depends on the built-in field and the stored charge density, we have:

$$-\frac{dn_{stored}}{dt} \propto \left\{ \begin{array}{l} TC(\xi_{trap}, E_{built}(n_{stored})) \cdot n_{stored} \\ +TC(\xi_{th}, E_{built}(n_{stored})) \cdot n_{th} + TC(\xi_{opt}, E_{built}(n_{stored})) \cdot n_{opt} \end{array} \right\} \quad (7)$$

where the first and second terms on the right-hand side are related to the leakage process without any external illumination. The first term is the direct tunneling due to built-in field from the traps of the stored charges, and the second term is due to thermally excited charges from the traps to the conduction band with energy ξ_{th} . In the third term, we consider the optical excitation of the stored charges with the photon energy $\hbar\omega$ and the excited electron energy $\xi_{opt} = \hbar\omega + \xi_{trap}$, which are related to the tunneling coefficient TC , while the stored charge density in the storage layer can also affect the built-in field that assists the tunneling process.

Supplementary Figure 6. The working principle of programming and erasing of bP-PPT device. (a) Band diagram of bP/AlO/HfO layer when 18 V top gate voltage is applied. Electrons from bP tunnel into HfO₂ layer and are trapped below the conduction level. (b) Band diagram of AHA charge storage layer with a charge density of $1.5 \times 10^{13} \text{ cm}^{-2}$ in the HfO₂ layer without top gate voltage. Trapped charges can optically be excited and removed from the charge storage layer. (c), (d) The band alignment changes with the trapped charged density with (c) and without (d) the top gate applied. Long-dashed line, short-dashed line, and straight line refer to charge density of 1.5, 0.5, and 0 ($\times 10^{13} \text{ cm}^{-2}$), respectively.

2. In page 3, author demonstrate the advantages of the AHA gate dielectric such as reliable and faster operation due to the utilized HfO₂ dielectric layer with a lower formed barrier height. How fast can this device work? The charge trapping process and retention performance rely on the barrier between HfO₂ and Al₂O₃, the barrier between Al₂O₃ and BP, as well as the thickness of the tunneling layer. Please give the comprehensive analysis.

Our response: We thank the reviewer for the suggestions. We have improved the S.I. Note 4, 5, and 6 with a theoretical analysis regarding the operation speed of the device. Following the reviewer's comments, we include the barrier height between HfO₂ and Al₂O₃, the barrier between Al₂O₃ and bP, and the thickness of

each layer in our theoretical model. Based on the analysis, we changed the description of our manuscript regarding the barrier height between bP and Al₂O₃ and introduced charge trap sites to emphasize the stability of the AHA charge storage layer. The charge injection speed from the bP-channel to the storage layer depends on the tunneling barrier height. The barrier height between bP and Al₂O₃ is 2.9 eV, lower than that between bP and SiO₂ (~3.5 eV), suggesting Al₂O₃ enables faster programming speed.

As for the charge de-trapping speed from the storage layer to the bP-channel, the tunneling barrier height between the trapped charges in HfO₂ and the Al₂O₃ plays an important role. Considering the high work functions (~5 eV) of the metallic charge storage layers used in conventional floating gate memory, the work function of 2.75 eV in the HfO₂ layer for the majority of trapped charge is beneficial for fast de-trapping and resetting the device. Based on the analysis, the electrical programming time of μs ~ms and optical programming time of ns~ μs can be achieved in our devices. The revised S.I. Notes 5 and 6 are shown below:

Supplementary Note 5. From equations (3), (4), and (5) in S.I. Note 4, the stored charges can screen the effective electric field in the tunneling layer. Hence, the FN tunneling current is suppressed with increasing density of the stored charge, which we calculate and plotted in Fig. S7a. The initial programming of the conductance of the bP-PPT channel requires 10s of ms gate pulse to saturate the charge density in the trapping layer, as shown in Fig. S7b. Different gate voltages and pulse time are considered in Fig. S7a and b, which result in different conductance states, as shown in Fig. 2C of the main text. Shorter pulses on the order of μs can be used to program the devices in smaller steps of conductance.

Supplementary Figure 7. The electrical programming speed (a) The FN tunneling versus the trapped charge density in the storage layer with different top gate voltage V_G . (b) Different charge densities in the storage layers depending on the pulse duration. Inset: Zoomed-in plot of (b) with shorter pulses.

Supplementary Note 6. For optical programming, we used optical pulses of ms duration in our experiment. But based on the theoretical analysis, the programming speed can be increased with ns~ μs optical pulses, depending on the pulse intensity and photon energy. Mechanisms of de-trapping of the stored charges in the HfO₂ layer include field-assisted tunneling, thermally excited charge tunneling, and optically excited charge tunneling as indicated in equation (7). Without the gate voltage, the tunneling coefficients for the first two mechanisms are negligible compared to the optically excited tunneling coefficient as shown in Fig. S8a. Because the trapped charges have to overcome the large tunneling barrier with Φ'_B , the charge

tunneling process can be activated with optical illumination, which reduces the effective barrier to $\Phi'_B - \hbar\omega$. The optically assisted tunneling process can be described as:

$$\begin{aligned} -\frac{dn_{stored}}{dt} &\propto TC(\xi_{opt}, E_{built}(n_{stored})) * n_{opt} \\ &= TC(\xi_{opt}, E_{built}(n_{stored})) * (G(P_{opt}) * t) \end{aligned} \quad (7)$$

where G is charge generation rate due to the optical illumination, which is assumed to be proportional to the optical power P_{opt} . To solve this nonlinear differential equation, we assume the conductance of the bP channel is in the range of 30 - 80 μS , which corresponds to the stored charge density of $4 - 12 \times 10^{12} \text{ cm}^{-2}$, as discussed in S.I. Note 3. Also, we approximated the analytical function of TC with an exponential function and only considered the trap site with 1.25 eV below the conduction band, where the largest density of oxygen vacancies in HfO_2 exists as reported with optical absorption spectra by Gritsenko, V. A., *et al.* (Ref. 12: “Electronic properties of hafnium oxide: A contribution from defects and traps”. *Physics Reports*, 613, 1-20. (2016)).

The resulting trapped charge density n_{stored} versus illumination time is plotted in Fig. S8b. The operation speed can be further reduced to the ns regime with higher photon energy and higher optical power. Considering our experimental conditions: the optical power on the bP-PPT device is 12 μW , the pulse width is varied from 1 ms to 200 ms, and photon energy is 1.6 eV (780 nm). We use equation (7) to calculate the optical pulse energy required to change the device from state $\#(n-1)$ to state $\#n$, which is plotted in Fig. S8c. The model shows a good agreement with the experimental results in Fig. S5a.

Supplementary Figure 8. (a) Tunneling coefficients depending on the stored charge density in the storage layer. (b) The conductance of bP-channel changes depending on the optical power and illumination time. (c) The calculated and experimental optical energy depending on the pulse number n , that changes the device from state $\#(n-1)$ to $\#n$.

3. What's the influence of the AHA thickness on the electrical and optical performance of the BP floating gate device?

Our response: The thickness of the AHA dielectric layer has a significant influence on the performance of the bP-PPT devices, as we have theoretically analyzed in the responses to questions 1 and 2.

From equations (1) and (2), the thicker the tunneling Al_2O_3 layer, the less the FN tunneling current for both charge injection from the bP channel to the HfO_2 charge storage layer and charge leakage from the HfO_2 to the bP channel. Therefore, a thicker Al_2O_3 layer will slow the programming speed but improve the retention time. The thickness of the HfO_2 trapping layer can influence the areal charge trap density (cm^{-2}) and the build-in field. A thicker HfO_2 layer will have a higher charge trap density due to increased charge trapping sites, increasing the modulation range of the channel conductance. However, thicker HfO_2 also reduces electrical programming speed due to the reduced field in the tunneling layer and therefore the tunneling current. Finally, for the blocking layer of Al_2O_3 , it needs to be thick enough to block the injected charges leaking from the trap layer during operation. However, too thick the blocking layer will also decrease the charge injection efficiency and limit the operation speed.

Therefore, there is a clear trade-off for the thickness of each layer in the AHA stack. Our device is optimized to achieve a long-enough retention time for the memory function, high charge injection efficiency for fast programming and operation speed, and device reliability and repeatability.

4. How to evaluate the degradation problem of this BP floating gate device? Because this hinders the practical application of the device.

Our response: We have evaluated the performance of the bP-PPT devices over a long period of time (up to 3 months) and included the results in the revised S.I. As shown in the figure below (also in Figure S11), the gate modulation, memory window, on-off ratio, and retention time of our devices remain consistent even 3 months after they are fabricated.

The excellent stability and longevity of the devices can be attributed to the protection by the AHA stack and our meticulous fabrication process that minimizes degradation and contamination of the bP. The 35 nm AHA ($\text{Al}_2\text{O}_3/\text{HfO}_2/\text{Al}_2\text{O}_3$) layers are deposited by atomic layer deposition (ALD), which effectively passivate and protect the bP. The result is consistent with the literature (Gamage, S., *et al.* ("Nanoscopy of black phosphorus degradation." *Advanced Materials Interfaces* 3, 12, 1600121 (2016)), which reported that 20 nm ALD alumina (Al_2O_3) can conformally coat the device and protect the thin bP flake from degradation for more than 90 days.

To further confirm the devices' stability during the device operations, we also conducted the endurance test that showed multi-state programming with excellent repeatability over 200 cycles. The result is shown in the figure below and in Figure S1c.

Supplementary Figure 1. (a) Time trace of state retention for two conductive states. (b) 3 bits (8 states) programming with electrical pulses. The first 18 V voltage pulse sets the device to the highest conductance state. Following depressive pulses of 20-ms with different voltage from -4 V to -12 V set the device to 8 different states (color coded). (c) Endurance test of the bP-PPT devices for 200 cycles of repeated procedure described in (b).

5. For Figure 2c, it just shows the retention time about 2000s, which is not consistent with the description in the page 4.

Our response: We thank the reviewer for pointing out this issue that may lead to a misunderstanding of the device retention time. We have revised the description in the manuscript to clarify. The result in Fig. 2c and the description on page 4 refer to the retention time of two different operation modes. The one with 2000s in Fig. 2c is for electrically programmed states with 8-states, the other with 1000 s referred to optically programmed states with 36-states. Increasing the number of states requires more strict requirements for distinguishing the programmed states. Therefore, the retention time on page 4 for 36 optical programmed states is shorter than that in Figure 2c for the electrical programmed 8 states.

6. Authors demonstrate that the device can realize 3 bits or even 5 bits conductance states. How to differentiate the different states? Is there some criterion?

Our response: We agree with the reviewer that a criterion is required to differentiate different states. The conductance states of our bP-PPT devices are determined with the following two criteria:

- (1) Each state is stable within a 99 % confidence interval (C.I.) (2.5 standard deviations) upon repeated programming of the desired states.
- (2) The separation between the states is more than 6 times of the standard deviation (6σ) so that each state can be clearly distinguished from the two adjacent states.

The figure below shows experimental results of well-separated conductance states of our devices that fulfill the two criteria. Here, we repeatably program the desired states #21, #20, #19 and #2, #1, #0, and analyze the statistics of the states' conductance values. Using the 99% C.I. criterion, we can conclude that the adjacent states are precisely set and clearly separated.

7. In Figure 3a, authors demonstrate that the array can act as both the optical frontend to receive and preprocess optical images and an electrical processor with in-memory computing to post-process the images. However, the connection between two applications (edge detection and image recognition) is not demonstrated clearly. So authors can extend the descriptions of the connections between two array applications more to help reader understand its significance.

Our response: We thank the reviewer for the insightful suggestions. We have added more detailed information about the measurement scheme in the revised S.I.

Currently, the receiving and pre-processing of the optical images and post-processing are performed with the same devices but in two steps. Our device first receives the image encoded in the telecom-band light and pre-processes the image with the programmed photoresponsivity. The result of this pre-processing is stored and electrically input to the device again for post-processing to perform the pattern recognition task. This two-step operation is because of the limited number of devices used in our experiment. In future edge computing image sensors, both the optical frontend (including receiving and pre-processing) and the image postprocessing stages can be implemented with large arrays of bP-PTT devices. Signal amplification circuits, also realized with 2D materials such as TMDCs, can be used to connect the two stages. We have added the above discussion in the revised manuscript.

8. The edge detection in Figure 3 is mainly based on the 2×2 bP-PPT array. How to evaluate the detection results?

Our response: We evaluate the edge detection result by comparing the measured values of each pixel to the simulated results. The ideal measurement case is expected to show a perfect linear correlation with the simulated results, as shown in the figure below and Fig. S13a. The correlation coefficient between the experimentally processed images and simulated images is plotted in Fig. S13b for the three demonstrated images, which are all better than 92%.

Supplementary Figure 13. Evaluation of the edge detection result by comparing the measured values of each pixel to the simulated results for all three pictures (MNIST handwritten digits, a husky dog, and a cameraman). (a) Experimental output photocurrent versus the simulated result, which shows linear correlation. (b) The correlation coefficients between the experimentally processed images and simulated results. All three pictures show the correlation coefficient over 92%.

REVIEWERS' COMMENTS

Reviewer #1 (Remarks to the Author):

The authors have addressed all my comments in positive spirit. I have found the manuscript improved, and I think it will be of broad interest to the journal readership. I would like to recommend that this manuscript be published.

Reviewer #2 (Remarks to the Author):

The authors have made a strong debate with a good scientific spirit. They have outlined some valid aspects of novelty however, I would like the authors to add a discussion in the paper that clarifies the novel elements for the reader. Although some of the points of difference are not crystal clear novelties in my opinion, I am satisfied with the overall argument presented. I believe this should be part of the manuscript perhaps at a relevant location in the introduction.

Reviewer #3 (Remarks to the Author):

The authors answer the questions very carefully and also highlight the novelty of the manuscript. It can be published in its current revised version.

The Authors' Response to Reviewers' Comments-2

Reviewer #1:

The authors have addressed all my comments in positive spirit. I have found the manuscript improved, and I think it will be of broad interest to the journal readership. I would like to recommend that this manuscript be published.

Our response: We thank the reviewer for the very positive and insightful comments on our work, and recommendation for the publication.

Response to Reviewer #2 :

The authors have made a strong debate with a good scientific spirit. They have outlined some valid aspects of novelty however, I would like the authors to add a discussion in the paper that clarifies the novel elements for the reader. Although some of the points of difference are not crystal clear novelties in my opinion, I am satisfied with the overall argument presented. I believe this should be part of the manuscript perhaps at a relevant location in the introduction.

Our response: We thank the reviewer for the insightful comments on clarifying our work's novelty. Our introduction has included a discussion about the advantages of in-sensor computing and references to previous studies using other 2D materials, and additional advantages of using bP for the extended spectral band to the IR regime.

Preprocessing the images within the sensors at the edge rather than in the cloud can largely alleviate the data streaming load to the servers, improving the bandwidth budget³⁵⁻³⁸ and reducing latency and power consumption. These advantages of edge computing have urged the development of optoelectronic edge sensors that combine vision-sensory and computational functionalities in the same devices^{8-10,19}, which recently have been demonstrated using 2D materials for visible/UV spectral imaging. Realizing such a scheme using bP will extend it to the infrared spectral range, enabling intelligent night vision and multispectral sensing.

To further emphasize the multi-functionality novelty of bP-PPT device, we have revised the introduction as follows:

The sensor can be programmed and read out both electrically and optically, enabling optoelectronic in-sensor computing, electronic in-memory computing, and optical remote programming, all in one device.

Reviewer #3 :

The authors answer the questions very carefully and also highlight the novelty of the manuscript. It can be published in its current revised version.

Our response: We thank the reviewer for the positive and insightful comments on our work.